# A Study on Initial Setting and Modulus of Elasticity of AAM Mortar Mixed with CSA Expansive Additive Using Ultrasonic Pulse Velocity

**DOI:** 10.3390/ma13194432

**Published:** 2020-10-05

**Authors:** Gum-Sung Ryu, Sung Choi, Kyung-Taek Koh, Gi-Hong Ahn, Hyeong-Yeol Kim, Young-Jun You

**Affiliations:** 1Department of Infrastructure Safety Research, Korea Institute of Civil Engineering and Building Technology, 283 Goyangdae-Ro, Ilsanseo-Gu, Goyang-Si 10223, Korea; ktgo@kict.re.kr (K.-T.K.); agh0530@kict.re.kr (G.-H.A.); hykim1@kict.re.kr (H.-Y.K.); yjyou@kict.re.kr (Y.-J.Y.); 2Department of Civil Engineering, KyungDong University 27, Gyeongdongdaehak-Ro, Yangju-Si 11458, Korea; csomy1113@kduniv.ac.kr

**Keywords:** alkali-activated material, CSA expansive additive, ultrasonic pulse velocity, setting time, modulus of elasticity

## Abstract

This study investigated the hardening process of alkali-activated material (AAM) mortar using calcium sulfoalumiante (CSA) expansive additive (CSA EA), which accelerates the initial reactivity of AAMs, and subsequent changes in ultrasonic pulse velocity (UPV). After the AAM mortar was mixed with three different contents of CSA EA, the setting and modulus of elasticity of the mortar at one day of age, which represent curing steps, were measured. In addition, UPV was used to analyze each curing step. The initial and final setting times of the AAM mortar could be predicted by analyzing the UPV results measured for 14 h. In addition, the dynamic modulus of elasticity calculated using the UPV results for 24 h showed a tendency similar to that of the static modulus of elasticity. The test results showed that the use of CSA EA accelerated the setting of the AAM mortar and increased the modulus of elasticity, and these results could be inferred using UPV. The proposed measurement method can be effective in evaluating the properties of a material that accelerates the initial reactivity.

## 1. Introduction

Alkali-activated materials (AAMs) are binders that accelerate the reaction of mineral admixtures, such as ground granulated blast furnace slag (GGBFS) and fly ash (FA), using strong alkali activators with high pH (>12). As AAMs are highly reactive, the hydrates of mineral admixtures are generated earlier. This lowers the fluidity of the mortar and causes setting and hardening at the same time. This process occurs quite suddenly [1,2,3,4]. The acceleration of the mortar setting has a significant impact on the physical properties of mortar at early ages [5,6,7,8,9], especially the initial shrinkage stress of the AAM mortar. According to the studies of many researchers, the shrinkage of mortar and concrete is evaluated after the occurrence of the final setting [10,11,12,13,14]. This is because the final setting means that the AAM matrix has been hardened and the deformation that occurs after the hardening of the AAM matrix acts as stress. Therefore, if the final setting of mortar that uses AAMs as binders is accelerated, shrinkage stress occurs earlier. This may cause significant initial shrinkage of the AAM mortar and high shrinkage stress. In addition, as the hydration reaction of AAMs is fast even after the final set, the strength and modulus of elasticity of the AAM mortar also significantly increase at early ages. The increase in the modulus of elasticity of the mortar causes a larger initial shrinkage stress. In general, additives, such as expansive additives (EAs) and shrinkage reducing agents, are used to control the shrinkage of mortar. These additives are effective in reducing shrinkage, but shrinkage stress can be evaluated differently if the setting and modulus of elasticity of mortar are considered [15]. In particular, when a calcium sulfoalumiante (CSA) expansive additive (CSA EA) is used, the formation of ettringite and calcium monosulfoaluminate at the initial stage can compensate for shrinkage, but it can also increase the modulus of elasticity and compressive strength [16,17,18,19]. If the setting is accelerated or the modulus of elasticity increases despite the reduction in shrinkage, the initial shrinkage stress of mortar can be evaluated to be larger [20]. Therefore, mortar and concrete that used AAMs require a comprehensive evaluation of the properties that can calculate the stress caused by deformation, such as setting, modulus of elasticity, and shrinkage at early ages.

In general, it is possible to measure the shrinkage of mortar before setting using an embedded strain gauge and a data logger. However, to measure the shrinkage stress of mortar, the modulus of elasticity at the time of the occurrence of shrinkage must also be considered.

The elastic modulus is determined from the load–displacement curve, as it is the slope of the curve. Nevertheless, the strength of concrete and mortar specimens is too low to be measured with mechanical tests, making it difficult to determine elastic modulus. In addition, as the strength and modulus of elasticity vary dramatically over time at early ages, the limited number of specimens to perform continuous monitoring may cause larger experimental errors [21,22,23]. The use of the ultrasonic pulse velocity (UPV), a type of non-destructive testing, can detect the physical properties of a specimen continuously and successively without causing damage. With respect to the analysis of the properties of concrete, UPV has been verified by many researchers [24,25,26,27]. Öztürk et al. detected the early hydration status of cement materials through ultrasonic reflectance measurements. They showed that changes in reflected waves responded well to various stages of hydration [28]. Voigt et al. conducted studies on changes in microstructure during cement hydration using ultrasonic reflection and transmission techniques [29,30].

In addition, the use of UPV makes it possible to quantitatively analyze the hydration process of concrete. In particular, many researchers have reported that the setting time can be predicted using changes in the UPV curve [31,32,33]. Reinhardt and Grosse developed an ultrasonic device for concrete quality control testing that can continuously observe the characteristics of concrete. The results obtained using this device reported that the initial setting time can be determined by UPV [34]. Krauß and Hariri reported that the degree of hydration of concrete at the initial stage can be analyzed using UPV and proposed a method to predict the setting time [35]. Belie et al. conducted research on the setting and hardening of shotcrete mortar, for which the hydration reaction of the binder is significantly fast. They found that shotcrete mortar was immediately hardened in the fluid state, and they could analyze the sudden setting and hardening process of the mortar using UPV [36].

The use of UPV makes it possible to analyze the overall hardening process of concrete and mortar, and UPV has been used to analyze the static and dynamic elasticity characteristics as well as the tensile and compressive strengths of concrete and mortar [35,37,38,39,40,41]. Rajagopalan et al. reported that they identified highly reliable concrete characteristics at early ages using the relationship between the UPV and compressive strength of concrete at early ages [42]. Anderson and Seals conducted research to predict the long-term compressive strength of concrete through non-destructive testing using UPV and proposed a non-destructive test method for predicting the long-term strength [43]. Abdel-Jawed and Afaneh investigated various factors in concrete that affect the ultrasonic pulse. They mentioned that the compressive strength characteristics over time according to the water/binder (W/B) ratio can be analyzed using UPV [44]. Trtnik et al. investigated material and mix properties that affect the UPV of concrete and verified the relationship between UPV and the static/dynamic modulus of elasticity based on the results [39].

There have been attempts of measuring UPV as a means of exploring the fresh state and setting properties of ordinary Portland cement (OPC)-based concrete and mortar, while it has been rarely applied to the binders that set very rapidly. Determination of elastic modulus after the setting has occurred is particularly important for controlling the shrinkage-induced stress and crack formation. This study attempted to evaluate the initial physical properties of the AAM mortar using CSA EA, which affects the setting and modulus of elasticity of mortar despite its shrinkage compensation effect. It was possible to find the initial and final setting times of the AAM mortar in the UPV inflection section by measuring the UPV for 48 h and to identify changes in its modulus of elasticity from the final setting to 48 h through the relationship between the UPV and dynamic modulus of elasticity. Based on this, the relationship between the setting and modulus of elasticity of the AAM mortar was analyzed according to the content of CSA EA, which is used for shrinkage reduction. 

## 2. Materials and Methods

### 2.1. Materials and Mixture Proportions of AAM Mortar

Table 1 shows the physical properties and chemical compositions of the ground granulated blast furnace slag (GGBFS), fly ash (FA), and calcium sulfoalumiante expansive additive (CSA EA) used in AAM mortar. GGBFS satisfied the third type of KS F2563 (density: >2.8 g/cm^3^, fineness: 4000–6000 cm^2^/g), and FA met the second type of KS L5405 (density: >1950 g/cm^3^, fineness: >3000 cm^2^/g). Both GGBFS and FA were produced by the company Sampyo in South Korea [45,46]. The CSA EA used in this study was POWER CSA TYPE from Denka in Japan. It was used to compensate for the shrinkage of the AAM mortar and was added on the basis of the binder mass. The main chemical components of GGBFS were CaO (41.9%), SiO_2_ (13.8%), and Al_2_O_3_ (4.9%). Its basicity coefficient (K_b_ = (CaO + MgO)/(SiO_2_ + Al_2_O_3_)) and hydration modulus (HM = (CaO + MgO + Al_2_O_3_)/SiO_2_) were 0.99 and 1.82, respectively. K_b_ was close to 1.0, which is a neutral value for ideal alkali activation, and HM was higher than 1.4, which is a value for excellent hydration reaction. FA was composed of SiO_2_ (56.8%), Al_2_O_3_ (22.8%), Fe_2_O_3_ (6.9%), and M_2_O(K_2_O+Na_2_O) (1.9%). The main components of CSA EA were lime, gypsum, and bauxite, and they were composed of CaO (34.6%), SiO (30.2%), and Al_2_O_3_ (24.2%). The alkali activator was used to accelerate the reaction of the binder. The alkali activator was in the form of a white powder with a molar ratio of 0.95. In addition, alkali activators are manufactured separately by adjusting the chemical components. The SiO_2_/Na_2_O ratio of the alkali activator used in this study was 0.92. The fine aggregate used was river sand with a density of 2.53, water absorptivity of 1.08, and fineness modulus of 2.77.

Table 2 summarizes the mix proportions of the AAM mortar. A binary blended binder in which GGBFS and FA were mixed at a ratio of 7:3 was used, and 24% alkali activator was used compared to the unit water content. The contents of CSA EA used were 0%, 2.5%, 5.0%, and 7.5% compared to the amount of the binder. The water/binder (W/B) ratio was 45.1%, and the sand/binder (S/B) ratio was 1.2. For the mixing method, the basic binder, alkali activator, and CSA EA were placed in a test container and dry mixing was performed for 30 s. Then, water was added, and mixing was performed for ten min at 150 rpm to produce paste. In addition, fine aggregate was added, and mixing was performed for 90 s at 300 rpm.

### 2.2. Test Methods

#### 2.2.1. Setting Time Test

To evaluate the setting characteristics of the AAM mortar, the mortar Vicat test was conducted in accordance with ASTM C191-18a [47]. The initial and final settings of the AAM mortar were determined using the penetration depth of the Vicat needle (diameter: 1.00 ± 0.05 mm, length: 50 mm or more) in a mortar for 30 s. The Vicat needle penetration tests were conducted every five min, and the time at which no trace of the Vicat needle was observed on the AAM mortar surface was determined as the final setting time.

#### 2.2.2. Compressive Strength Test

The compressive strength test was conducted in accordance with ASTM C109-16a [48]. The diameter and height of the cylindrical specimens were 100 and 200 mm, respectively, and the tests were conducted at curing ages of 1, 7, and 28 days. The average compressive strengths of the three specimens were used. The specimens were cured in a constant temperature and humidity chamber at a temperature of 23 ± 2 °C and a relative humidity of 90 ± 2%.

#### 2.2.3. Modulus of Elasticity Test

The concrete modulus of elasticity test was conducted in accordance with ASTM C469M-14 [49]. The diameter and height of the cylindrical specimens were 100 and 200 mm, respectively. Two strain gauges were attached to the side of each specimen to measure the modulus of elasticity. A load of up to 40% of the ultimate load was applied at a rate of 0.25 MPa/s, and the modulus of elasticity was calculated through regression based on the interpolation of strain measurements for the load. The modulus of elasticity test results was obtained by averaging the modulus of elasticity measurement results of three specimens. The specimens were cured in a constant temperature and humidity chamber at a temperature of 23 ± 2 °C and a relative humidity of 90 ± 2%.

#### 2.2.4. Ultrasonic Pulse Value (UPV) Test

The system used for monitoring the UPV in the AAM mortar specimens is shown in Figure 1. A hole was made in the Styrofoam container, and an oscillator and a receiver were placed so that they could face each other at a distance of 30 mm. After filling the container with mortar, the transit time of ultrasonic waves from the oscillator to the receiver was measured every 30 s. The Pundit-2 model from PROCEQ was used to measure the UPV. UPV (V_c_) was obtained by dividing the length of the specimen (L) by the transit time (T), and it has the following relationship with the dynamic modulus of elasticity (E_d_) and density (*ρ*) [50,51,52].
(1)$$V_{c}=\frac{L}{T}=\sqrt{\frac{E_{d}}{\rho }}$$

Figure 2 shows the typical evolution curve of the UPV of the AAM mortar, which was altered with three sections. According to the curve, points A and B can be determined in terms of altered time points. The UPV begins to increase at point A, and it begins to slow down and converges to a certain value at point B [53,54,55].

## 3. Results and Discussion

### 3.1. Setting Time

The setting of AAM mortar is affected by the conditions of the alkali activator, and it is generally faster than that of ordinary Portland cement (OPC) [56,57]. Table 3 shows the setting time of the AAM mortar according to the content of EA. For 0 EA without EA, the initial setting time (IST) was 101 min and the final setting time (FST) was 292 min. As for paste (w/c = 0.5) using ordinary Portland cement (OPC), the initial setting is within 6 to 7 h according to the literature [58]. Under the same mixing conditions, the AAM mortar exhibited a faster setting compared to mortar that used cement as a binder. This is because the alkali activator stimulated the binder quite early [59].

The setting time was accelerated as the CSA EA content increased. For 7.5 EA with the highest CSA EA content, the initial setting time and final setting time were 55 and 171 min, respectively, which were 45.4% and 41.4% shorter compared to those of 0 EA. When a CaO-based CSA EA is used in mortar, Ca^2+^ ions diffuse and generate hydrates (Ca(OH)_2_). The setting characteristic of OPC is highly associated with the hydration of C_3_S (Ca_3_SiO_5_, alite). An OPC paste reaches its initial setting state once C_3_S has been sufficiently hydrated, and the final setting occurs when a sufficient amount of C-S-H has precipitated. On the other hand, the main constituent of CSA expansive additives is C_4_A_3_s (Ye’elimite), which dissolves in contact with water and forms ettringite (Ca_6_Al_2_(SO_4_)_3_(OH)_12_·26H_2_O) at a much faster rate than C_3_S does.

This action, which is similar to the alkali activation reaction of AAM, accelerates the reaction of AAM by increasing the dissolution of Ca^2+^ ions in the mixing water, thereby further accelerating setting [59,60]. 

### 3.2. Compressive Strength and Modulus of Elasticity

When a CSA EA is used in mortar, the mortar temporarily expands at the initial stage owing to the hydrates of EA, and its internal structure becomes dense. This increases the strength and modulus of elasticity of the mortar [1]. Table 3 shows the compressive strength and modulus of elasticity of AAM mortar mixed with CSA EA. 

The average compressive strength of the three AAM mortars exceeded 40 MPa, which was the target strength set during the mix design. The AAM mortar without EA (0 EA) exhibited high initial strength, with a compressive strength of 3.2 MPa and modulus of elasticity of 1.2 GPa at 1 day of age. However, for the AAM mortar mixed with CSA EA, the compressive strength was higher than 5 MPa, and the modulus of elasticity was higher than 2 GPa at 1 day of age, which were 63–87% and 76.3–103.5% higher, respectively, compared to those of 0 EA. AAM mortar containing CSA EA exhibited higher compressive strength values at 28 days of age compared to 0 EA. At one day of age, the strength and modulus of elasticity increased as the EA content increased. However, at 28 days of age, the compressive strength of 5.0 EA was the highest (49.6 MPa), and the compressive strength of 7.5 EA was 47.3 MPa, which was lower than that of 2.5 EA.

When the modulus of elasticity of the AAM mortar at 28 days of age was compared, the modulus of elasticity of the AAM mortar containing CSA EA was higher than that of 0 EA, but there was no clear improvement in the modulus of elasticity due to EA. This is because the modulus of elasticity was not significantly affected by the binder, as it was also affected by factors other than the cement matrix, such as fine aggregate.

### 3.3. Ultrasonic Pulse Velocity

UPV can measure changes in the internal structure of mortar caused by a hydration reaction without physical damage using the speed of sound waves, and it is possible to identify sudden changes in the internal structure of mortar through the monitoring of UPV. According to Lee et al., the UPV of mortar and concrete has three different sections [61]. The initial fluid state section (step 1) represents the unhardened state of the mortar. In this instance, the UPV exhibits a level similar to that of the fluid state and ranges from 300 to 500 m/s. In step 2, pores are filled with the hydration products of cement, and moisture unsaturated porous solid structures are connected, thereby increasing the UPV. In the step 3 section, high UPV is maintained because the hydrates are connected to each other, and mortar begins to develop strength owing to the curing process. Micropores are filled with hydration products, and UPV continuously increases, even though it does not increase as rapidly as in step 2.

Figure 3 shows the UPV curves of the AAM mortar according to the EA content. The UPV of the AAM mortar incorporating CSA EA was 348–412 m/s, corresponding to the values typically shown by a mixture in the fluid state. An increase in the UPV value was observed after 0.5 h by 7.5 EA, which incorporated the highest dosage of CSA EA and was the first to enter step 2, followed by 2.5 EA and 0 EA after 1.5 h. It was observed that the UPV rapidly increased in step 2, while its increasing rate decreased after 3–8 h, reaching step 3. Similarly, the time taken to reach step 3 was the shortest in 7.5 EA. The rate at which the UPV increased tended to gradually decrease 48 h after the specimens reached step 3.

The curves show shapes similar to those of the UPV curves of other researchers, but there are some differences. In a study by Lee et al. [61], the duration of step 1 for OPC mortar was 6–10 h, whereas that for the AAM mortar was 0.5–1.5 h, suggesting that the duration of step 1 was much shorter for AAM. In addition, while step 2 was reached after a sudden increase from step 1 for the UPV curves of the AAM mortar, the UPV curves of the OPC mortar continuously increased without clear boundary points to reach step 2. This is because the rapid hydration reaction occurred as AAM mortar-stimulated GGBFS using the alkali activator. In addition, CSA EA further reduced step 1 as it accelerated the hydration reaction. Lee et al. [61] suggested that OPC mortar reaches step 3 after 16–20 h, while it was shown that AAM took 3–8 h to reach step 3, showing a much faster rate than OPC mortar. This is because the rapid reaction of AAM mortar occurred as described above. The AAM mortar containing EA exhibited a higher UPV at the same age. This is due to the generation of ettringite, which is the main hydration product of EA.

### 3.4. Setting Time in UPV Test

The UPV curve can reflect changes in the internal structure of mortar caused by the hydration reaction. Pessiki and Carino et al. analyzed the relationship between UPV and setting or strength using the UPV curve [62]. According to a report by Lee et al., the analysis of the relationship between the UPV and the setting of concrete revealed that UPV increased when setting began. It was confirmed that initial and final settings occurred when UPV reached certain values [61,63,64].

The UPV measurement for AAM according to each step defined in Figure 2 is analyzed in Figure 4. As can be seen from the figure, the UPV curves are divided into three sections. According to a report by Lee, the intersections of the straight lines created by connecting UPV values in each section are related to setting. As the UPV curves of the AAM mortar have three different sections, straight lines were created in each section by connecting the data, and the intersections of the lines were compared with the setting time by the ASTM C191-18a test method.

Step 1 lasted for approximately 60 min (30 min for EA7.5), and the UPV results were used to create a straight line (black curve in Figure 4). The UPV sharply increased afterwards, and step 2 began approximately 90–120 min after mixing. In this instance, the use of EA advanced the time at which step 2 began. During step 2, the UPV results for an hour were used to create a straight line (blue curve in Figure 4). Step 3 began approximately eight hours after mixing. Mortar mixtures tend to reach step 3 within 3–8 h after setting. The point at which the UPV converges has been determined by regression analysis (blue curve in Figure 4).

To determine the setting time of the AAM mortar using the UPV curves, the intersections of the straight lines created in each section of the curves were examined. The intersection of the straight lines of the step 1 and step 2 sections was defined as the initial setting, and that of the straight lines of the step 2 and step 3 sections was defined as the final setting. The analysis results showed that the initial setting time of 0 EA was 102 min, and its final setting time was 294 min. In the same manner, the initial setting and final setting of 2.5, 5.0, and 7.5 EA are shown in each graph. The initial and final setting times determined using UPV became shorter as the EA content increased. Table 4 shows the differences between the initial and final setting times determined by the UPV curves and those determined by ASTM C 191-18a. The difference in initial setting time ranged from 1 to 14 min, and that in the final setting ranged from 1 to 10 min. In particular, when the EA content was 2.5% or less, the difference in setting time was 3 min or less, resulting in no significant difference in the results of the two test methods.

### 3.5. Elasticity in UPV Test

The dynamic modulus of elasticity of mortar can be obtained through an empirical equation if dynamic characteristics, such as UPV, are used.
(2)$$E_{d}=\rho V_{c}^{2}$$
where *E_d_* is the dynamic modulus of elasticity (GPa), *ρ* is the density of the specimen (ton/m^3^), and *V_c_* is UPV (m/s). Table 5 shows the UPV results of the AAM mortar at one day of age, the E_d_ values obtained using the results, and the static modulus of elasticity (E_s_) values at one day of age obtained through an experiment. The density of the AAM mortar used was 2.4 ton/m^3^.

As the EA content increased at 1 day of age, both E_d_ and E_s_ increased. As the content of CSA EA increased to 2.5, 5.0, and 7.5%, E_s_ increased by 23.5, 32.1, and 45.9% compared to that of 0 EA and E_d_ increased by 29.1, 37.6, and 49.1%. Here, the difference between the increase rates of E_d_ and E_s_ of the AAM mortar that used CSA EA based on 0 EA ranged from 3.2% to 5.6%, indicating that E_d_ and E_s_ showed relatively similar tendencies. This means that E_s_ can be indirectly calculated through UPV because E_d_ is obtained using the equation of UPV. This approach has been mainly used to analyze the characteristics of rocks, and the correlation between the static and dynamic characteristics of rocks has been verified by several researchers [65,66,67,68,69].

In general, the dynamic modulus of elasticity tends to be higher than the static modulus of elasticity [65,70,71,72]. This can be because an increase in the crack and pore volume in concrete and mortar decreases their static modulus of elasticity to a larger extent [65,73]. In the test results, E_d_ by UPV exhibited higher results than E_s_. It was found that E_d_ was 62.4–55.4% higher than E_s_.

Table 6 shows the static modulus of elasticity of the AAM mortar at the final setting time obtained using UPV. The UPV at the accurate final setting time was unknown because it was measured every 30 min. Therefore, the dynamic modulus of elasticity was calculated using UPV at a time point similar to the final setting time. At the final setting time of the AAM mortar, UPV exhibited similar values, ranging from 2039.9 to 2205.0 m/s. Based on this, the dynamic modulus of elasticity was measured to be between 0.98 and 1.14 GPa. The AAM mortar developed stiffness as it was cured, and UPV and the dynamic modulus of elasticity were found to be constant at the final setting time. They were similar regardless of EA content.

## 4. Conclusions

In this study, the initial hardening characteristics of alkali-activated material (AAM) mortar were analyzed using the four levels of CSA expansive additive (CSA EA) content. For the analysis of the initial hardening characteristics using ultrasonic pulse velocity (UPV), the results of the Vicat setting test, a conventional test method, were compared with those of the modulus of elasticity test. The main observations and findings of this study can be summarized as follows:
CSA EA is used to compensate for initial shrinkage, but it accelerates the initial reactivity and shortens the setting time, thereby causing the final setting, which is the basis of shrinkage stress. The dynamic modulus of elasticity of the AAM mortar obtained using UPV at the final setting time was approximately 1 GPa, which was constant regardless of the CSA EA content. As the modulus of elasticity of the AAM mortar increases over time, CSA EA that accelerates the final setting may cause high shrinkage stress at early ages.The compressive strength of the AAM mortar tended to increase with the CSA EA up to the dosage level of 5.0%, whereas it decreased at a dosage of 7.5%. Therefore, it is concluded that the maximum dosage of CSA EA should be limited to 5.0%, considering its effect on the compressive strength development.The initial and final setting times obtained by the UPV test were similar to those obtained by the Vicat test. The dynamic modulus of elasticity derived through UPV increased as the CSA EA content increased, and it exhibited increase rates similar to those of the static modulus of elasticity. Therefore, UPV can reflect the curing, initial setting, and final setting processes of AAM mortar caused by the initial alkali activation reaction, and it is possible to evaluate the degree of curing of AAM mortar through the dynamic modulus of elasticity.The use of a UPV monitoring system makes it possible to continuously observe the curing process of mortar and to identify changes in the stiffness of AAM mortar using the dynamic modulus of elasticity at early ages when it is difficult to conduct the static modulus of elasticity test. These data are expected to be useful in evaluating the initial shrinkage stress of AAM mortars.In addition, the relationship between UPV and the static modulus of elasticity was established through various tests according to the material properties. Based on this relationship, it will be possible to evaluate the stress acting on mortar due to the displacement caused by various causes at early ages.


## Figures and Tables

**Figure 1 materials-13-04432-f001:**
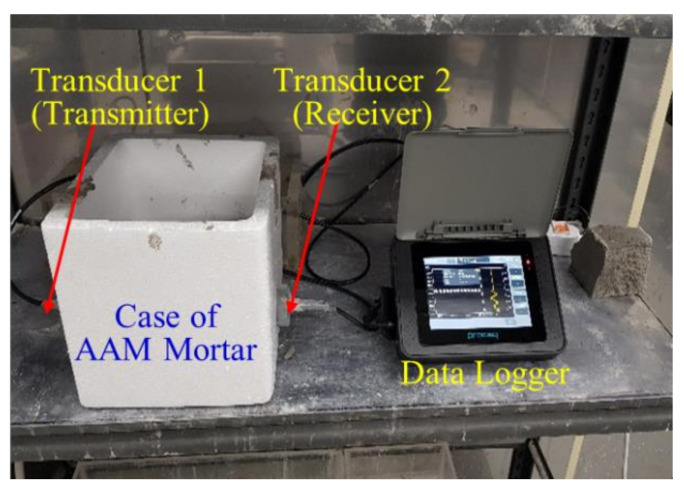
Schematic of ultrasonic pulse velocity (UPV) monitoring.

**Figure 2 materials-13-04432-f002:**
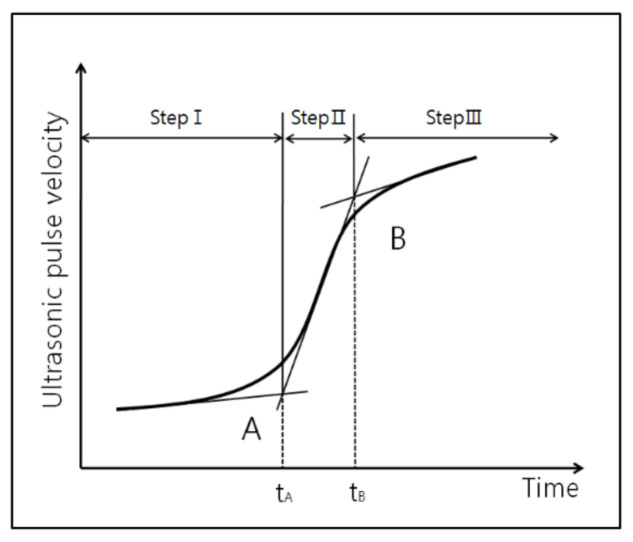
Typical evolution curve of ultrasonic pulse velocity.

**Figure 3 materials-13-04432-f003:**
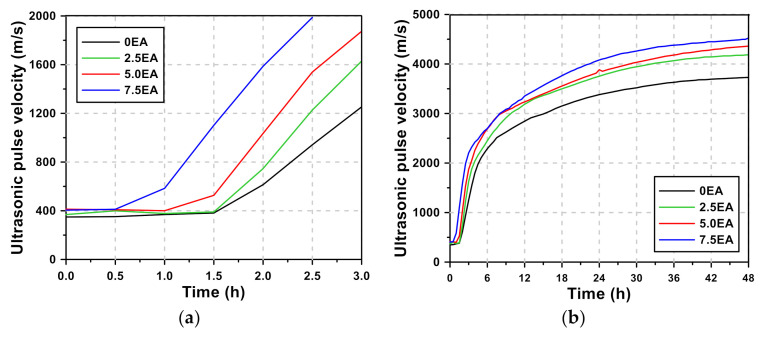
AAM mortar of ultrasonic pulse velocity curve: (**a**) Time: 3 h, (**b**) Time: 48 h.

**Figure 4 materials-13-04432-f004:**
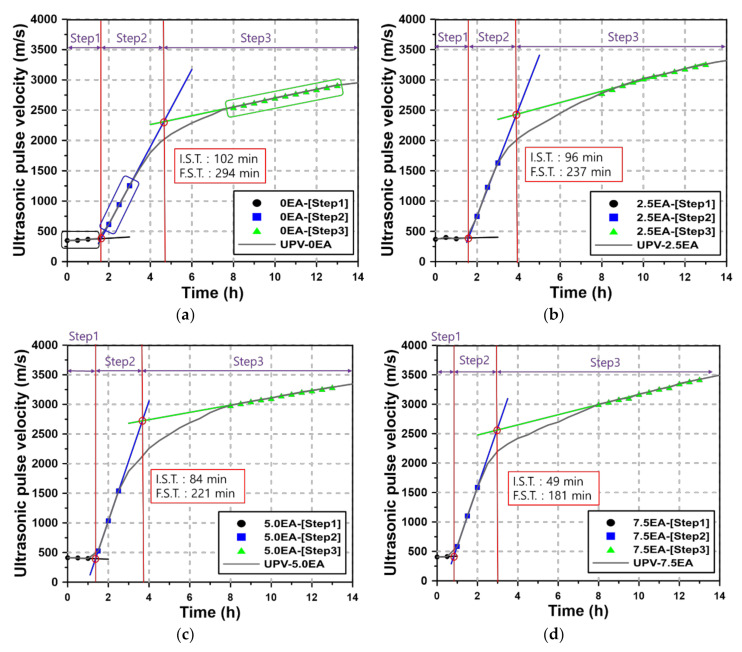
UPV curves of AAM mortar: (**a**) 0 EA, (**b**) 2.5 EA, (**c**) 5.0 EA, (**d**) 7.5 EA.

**Table 1 materials-13-04432-t001:** Physical properties and chemical composition of alkali-activated material (AAM) binder and calcium sulfoalumiante expansive additive (CSA EA).

Type	Density (g/cm^3^)	Blaine Specific Surface	CaO	SiO_2_	Al_2_O_3_	Fe_2_O_3_	SO_3_	MgO	K_2_O	Na_2_O
		(cm^2^/g)								
GGBFS	2.9	4680	43.4	34.6	14.3	0.6	5.0	5.1	0.5	0.2
FA	2.2	3220	3.5	56.8	22.8	6.9	0.5	1.8	1.1	0.8
CSA expansive additive	2.9	3750	36.4	30.2	24.2	1.7	5.3	1.4	0.5	0.3
Alkali activator	1.0	-	-	46.2	-	-	-	-	-	50.2

**Table 2 materials-13-04432-t002:** Mix proportions of alkali-activated material (AAM) mortar.

Type	Water	Binder	Activator	Sand	CSA
	(g)	(g)	(g)	(g)	(g)
0 EA	0.451	1.000	0.108	1.200	0.000
2.5 EA	0.451	1.000	0.108	1.200	0.250
5.0 EA	0.451	1.000	0.108	1.200	0.500
7.5 EA	0.451	1.000	0.108	1.200	0.750

**Table 3 materials-13-04432-t003:** Setting time and strength characteristics of the AAM mortar mixed with EA.

Type	Setting Time		Compressive Strength		Modulus of Elasticity	
	(min)		(MPa)		(E_s_, GPa)	
	IST	FST	1D	28D	1D	28D
0 EA	101	292	3.2	43.6	1.7	19.3
2.5 EA	93	238	5.2	49.1	2.1	20.1
5.0 EA	70	223	5.9	49.6	2.3	20.2
7.5 EA	55	171	5.7	47.3	2.5	20.2

**Table 4 materials-13-04432-t004:** Differences in setting time between the ASTM C 191-18a test method and UPV.

Type	ASTM C 191-18a—UPV (min)			
	0 EA	2.5 EA	5.0 EA	7.5 EA
IST *	1	3	14	6
FST **	2	1	2	10

IST *: Initial setting time; FST **: Final setting time.

**Table 5 materials-13-04432-t005:** UPV and dynamic and static moduli of elasticity of the AAM mortar at one day of age.

Type	By UPV Test (1D)			By ASTM C469M-14 (1D)		E_d_/E_s_
	UPV (m/s)	E_d_ (GPa)	Increase Rate	E_s_ (GPa)	Increase Rate	
			(0 EA)		(0 EA)	
0 E	3381.9	2.68	100.0	1.65	100.0	1.624
2.5 E	3753.5	3.31	123.5	2.13	129.1	1.554
5.0 E	3883.9	3.54	132.1	2.27	137.6	1.560
7.5 E	4081.1	3.91	145.9	2.46	149.1	1.589

**Table 6 materials-13-04432-t006:** UPV and dynamic modulus of elasticity at the final setting time of the AAM mortar.

Type	By ASTM C 191-18a	By UPV Test		
	Final Setting (h:m)	Time	UPV(m/s)	E_s_ (GPa)
0 EA	4:52	5:00	2105.1	1.04
2.5 EA	3:58	4:00	2039.9	0.98
5.0 EA	3:43	3:30	2056.6	0.99
7.5 EA	2:51	3:00	2205.0	1.14

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
