# Peer review of "A Study on Initial Setting and Modulus of Elasticity of AAM Mortar Mixed with CSA Expansive Additive Using Ultrasonic Pulse Velocity"

_materials, 2020, doi:10.3390/ma13194432_

Round 1

Reviewer 1 Report

In this paper, the initial setting and modulus of elasticity of AAM mortar mixed with three different contents of CSA EAs using UPV were studied. It was proved that the addition of CSA EA accelerated the setting of AAM mortar and increased the modulus of elasticity. Moreover, the suitability of non-destructive methods for the investigation of concrete structures was confirmed.

Dear authors, your paper is in my opinion well written and its scientific soundness is recognized. However, minor modifications are required, I suggest the following corrections:

Line 112: The abbreviation of GGBFS and FA should be described before the abbreviated version is used in the text.

Line 113 – 114: What is KS F2563 and KS L5405? Do you mean national standards? Are the requirements of these standards similar to the demands of some international standards? If you state that the requirements of KS .. were satisfied, what exactly do you mean? Size of granulate, density etc.

Author Response

Point 1: The abbreviation of GGBFS and FA should be described before the abbreviated version is used in the text. (Line 112)

Response 1: In accordance with the reviewer’s comment, the definitions of abbreviations GGBFS and FA have been provided in Line 38.

Point 2: What is KS F2563 and KS L5405? Do you mean national standards? Are the requirements of these standards similar to the demands of international standards? If you state that the requirements of the KS .. were satisfied, what exactly do you mean? Size of granulate, density, etc. (Lines 113–114)

Response 2: Because  the materials used in this study are generated in Korea, they are classified according to the Korean Standard. The GGBFS used here can be classified as grade 80, as specified in ASTM C 989, and FA can be classified as Class F, as specified in ASTM 618. Nevertheless, these materials are classified in accordance with KS F 2563 and KS L 5405, owing to their origins. To aid the readers, we have added a brief explanation of the relevant Korean Standard specifications.

 [A1]The word 'since' is generally used to convey a sense of time. For example, 'I have been working here since 2000.' The words 'because' or 'as,' on the other hand, are generally used to explain or provide a reason for something. For example, 'I enjoy working here because/as I find my role challenging.'

Reviewer 2 Report

A Study on Initial Setting and Modulus of Elasticity of AAM Mortar Mixed with CSA Expansive Additive Using Ultrasonic Pulse Velocity

Gum-Sung Ryu, Sung Choi, Kyeong-Taek Koh, Gi-Hong An, Hyeong-Yeol Kim, and Young-Jun You

The paper is about the evaluation of the use of UPV for the investigation of physical and mechanical properties of AAM mortar during curing time. AAM mortars are interesting novel materials worth of investigation.

The reviewer suggests to describe the original contribution of the paper and the improvement with respect to similar researches present in literature.

Figure 1 is missing. Please check the style of figures’ captions.

The reviewer suggests to reorganize the section, moving the parts about the descriptions and the literature considerations from the “results” sections.

Please add the comments of the results, also with theoretical explanations: now the paper appears similar to a report of experimental tests.

Also comparison is reported only for 1 day of curing and after setting time: please add further steps and comment.

Further comments are highly recommended, above all about the original contribution, with respect to literature and the applications of the results of the research. Which is the best material? What is the novelty proposed in this paper with respect to the scientific panorama? What is the real convenience of using UPV?

Further recommendations are reported in the attached file.

Reviewer 3 Report

This is a very good paper that is well written and reports a useful study of the effect of a CSA expansive additive on the setting of alkali-activated mortar using ultrasonic pulse velocity measurements.  These measurements allow the setting process to be studied in full without damaging the sample, and enable to initial and final setting times to be determined.  Results are reported on the specific effects of the expansive additive, as well as allowing the test method to be compared with more conventional destructive methods of studying the change in mechanical properties with setting.  Overall, this paper merits publication with only minimal editorial changes.  These are:

Line 238: Replace "... quite earlier compared to..." with "...sooner than..."

Line 344: the word "test" should be plural, i.e. "tests".

Author Response

Point 1: Replace "... quite earlier compared to..." with "...sooner than..." (Line 238)

Response 1: The relevant sentence has been revised in accordance with the reviewer’s comment.

Point 2: the word "test" should be plural, i.e. "tests" (Line 344)

Response 2: The relevant sentence has been revised in accordance with the reviewer’s comment.

Round 2

Reviewer 2 Report

Second revision.

The authors have addressed most of the previous suggestions of the reviewer.

The paper has been improved. Only few suggestions in the attached revised manuscript, about information to add in the text. Then, the paper'll be ready for publication.

Author Response

Point 1: Yes, slope, but where in the curve? Tangent? Or secant? In which point?

Response 1: the elastic modulus is the secant modulus, which is equal to the slope of the secant between the origin and a point on the stress-strain curve.

Point 2: please describe the different features in the figure, adding text and arrows

Response 2: In accordance with the reviewer’s comment, we have added the figure1 in the revised manuscript as follows (line 176)

Point 3: Please describe better the target. If it it the average, please add in the text

Response 3: In accordance with the reviewer’s comment, we have added the test method in the revised manuscript as follows (line 206)
